# Explaining the Space of SSP Policies via Policy-Property Dependencies: Complexity, Algorithms, and Relation to Multi-Objective Planning

**Primary Keywords:** *None*

## Abstract

Stochastic shortest path (SSP) problems are a common framework for planning under uncertainty. However, the reactive structure of their solution policies is typically not easily comprehensible by an end-user, while planners neither justify the reasons behind their choice of a particular policy over others. To strengthen confidence in the planner's decision-making, recent work in classical planning has introduced a framework for explaining to the user the possible solution space in terms of necessary trade-offs between user-provided plan properties. Here, we extend this framework to SSPs. We introduce a notion of policy properties taking into account action-outcome uncertainty. We analyze formally the computational problem of identifying the exclusion relationships between policy properties, showing that this problem is in fact harder than SSP planning in a complexity theoretical sense. We show that all the relationships can be identified through a series of heuristic searches, which, if ordered in a clever way, yields an anytime algorithm. Further, we introduce an alternative method, which leverages a connection to multi-objective probabilistic planning to move all the computational burden to a pre-process. Finally, we explore empirically the feasibility of the proposed explanation methodology on a range of adapted IPPC benchmarks.

## 1 Introduction

Stochastic shortest path (SSP) problems (Bertsekas and Tsitsiklis 1991) are a common framework for planning under action-outcome uncertainty. SSP solutions take into account arising contingencies by means of policies deciding what action to take next as a function of what happened in the past. This complex structure, however, leads to an astronomically large space $\Pi$ of different possible solutions, a direct overview of which is beyond the capability of actual end-users of planning technology. At the same time, current SSP solution methods lack functionality for providing reasons behind the choice of a particular policy over others.

Krarup et al.'s (2021) introduced a generic iterative planning framework aiming at helping a user to interactively explore and comprehend the solution space $\Pi$. The goal of the user-planner interactions is to arrive at a solution that is in line with the user's preferences. Eifler et al. (2020a,b) proposed an instantiation of this framework for classical planning. The user-planner interactions are based on a set of user-provided **plan properties** $\Omega$, i.e. Boolean functions on

plans. The plan space is explained by identifying trade-offs between the given properties, i.e., the relationships between property subsets $\Phi$ and $\Psi$ that are not simultaneously achievable. This gives rise to a *contrastive* question-answer kind of dialog (Miller 2019), where the user specifies preferences $\Phi \subseteq \Omega$ and is then confronted with the implied exclusion of other property subsets $\Psi$. This has proved highly beneficial in strengthening the user's understanding of the solution space (Eifler et al. 2022).

While explaining solutions to probabilistic planning models is by no means a novel topic, most prior work deals with the explanation of *individual* solutions. With the recognition of the importance of explainable AI, the sub-field of explainable reinforcement learning is increasingly gaining attention, yet its primary focus lies in tracing a policy's decision back to input features (Milani et al. 2023). Topin and Veloso (2019) attempted to summarize a single policy via state abstraction. Juozapaitis et al. (2019) and Khan, Poupart, and Black (2009) explain single action-decisions of a given policy by contrasting them to all available options by exposing trade-offs between different reward or cost objectives.

Building on Eifler et al.'s (2020a) explanation methodology, we propose a novel approach to explaining the **global** space of all SSP policies, or a **local** user-chosen subset. We introduce a notion of **policy properties**, taking into account action-outcome uncertainty, and formally study the computational problem of identifying their exclusion relationships. In order to compute the explanations, we devise two approaches that relate to constrained SSPs (Altman 1999) and multi-objective SSPs (Chen, Trevizan, and Thiébaux 2023), respectively. We show that both approaches can be efficiently realized by adapting the heuristic search algorithm I-DUAL (Trevizan et al. 2016), which embeds occupation-measure heuristics for effective guidance (Trevizan, Thiébaux, and Haslum 2017). Finally, we explore the feasibility of the proposed explanation methodology empirically on a range of adapted IPPC and new benchmarks.

## 2 Background

For a finite set $X$, we refer with $\Delta(X)$ to the set of all probability distributions over $X$. For $\delta \in \Delta(X)$, $supp(\delta) := \{x \in X \mid \delta(x) > 0\}$ gives the **support** of $\delta$.

## 2.1 Probabilistic Planning

We consider probabilistic planning tasks in a SAS+-like notation (Bäckström and Nebel 1995). A **planning task** is given by a tuple $\tau = \langle \mathcal{V}, \mathcal{A}, s_0, \mathcal{G}, c \rangle$, where $\mathcal{V}$ is a finite set of **state variables**, each $v \in \mathcal{V}$ having a finite **domain** $\mathcal{D}_v$; $\mathcal{A}$ is a finite set of **actions**; $s_0$ is the **initial state** (a complete variable assignment); $\mathcal{G}$ is a conjunctive **goal** (a variable assignment); and $c: \mathcal{A} \mapsto \mathbb{R}_0^+$ is the **cost function**. Pairs $\langle v, d \rangle$ of variables $v \in \mathcal{V}$ and values $d \in \mathcal{D}_v$ are called **facts**. The set of all facts is denoted with $\mathcal{F}$. The **states** of $\tau$ are the complete variable assignments. The set of all states is denoted with $\mathcal{S}$. For a (partial) variable assignment $G$, we denote with $\mathcal{S}_G := \{ s \in \mathcal{S} \mid G \subseteq s \}$ the subset of states *satisfying* $G$. Each action $a$ is associated with a **precondition** $pre_a$ (a variable assignment) and a probability distribution $Out_a$ over **probabilistic outcomes** (variable assignments). An action $a$ is **applicable** in a state $s$ if $s \in \mathcal{S}_{pre_a}$. The set of all actions applicable in $s$ is denoted with $\mathcal{A}(s)$. The state resulting from an action outcome $o \in supp(Out_a)$ is $s\llbracket o \rrbracket[v] = o[v]$ if $o[v]$ is defined, else $s\llbracket o \rrbracket[v] = s[v]$.

A task $\tau$ induces the **stochastic shortest path (SSP) problem** $\Theta_\tau = \langle \mathcal{S}, \mathcal{A}, \mathcal{T}, s_0, \mathcal{S}_\mathcal{G}, c \rangle$ with same states, actions, initial state, goal states, and cost function, and the **transition probability function** $\mathcal{T}: \mathcal{S} \times \mathcal{A} \mapsto \Delta(\mathcal{S})$. Unless important, we treat $\tau$ and $\Theta_\tau$ interchangeably.

We are interested in **deterministic policies** (policies for short), which are (partial) functions $\pi: \mathcal{S} \rightharpoonup \mathcal{A}$ such that $\pi(s) \in \mathcal{A}(s)$ if $\pi(s)$ is defined. For a policy $\pi$, we write $\pi(s) = \bot$ if $\pi(s)$ is not defined. The **terminal states** $\mathcal{S}_\bot^\pi \subseteq \mathcal{S}$ under $\pi$ are those $s$ where $\pi(s) = \bot$. A state $s'$ is **reachable** from a state $s$ via $\pi$ if there are states $s_1, \dots, s_n$ with $\pi(s_{i-1}) \neq \bot$ and $\mathcal{T}(s_{i-1}, \pi(s_{i-1}), s_i) > 0$, for all $1 < i \leq n$, such that $s_1 = s$ and $s_n = s'$. The set of all states reachable from $s$ via $\pi$ is denoted with $\mathcal{S}^\pi(s)$; the reachable terminal states with $\mathcal{S}_\bot^\pi(s)$.

Let $G$ be a variable assignment. The $G$-**reachability probabilities** of a policy $\pi$ is denoted with $P^{G,\pi}$. The $\mathcal{G}$-reachability probabilities are also called **goal probabilities**. We say that a policy $\pi$ is $s$-**proper** if $P^{\mathcal{G},\pi}(s) = 1$. We denote the set of all $s$-proper policies with $\Pi(s)$. The **expected cost** of a policy $\pi$ under the cost function $c$ is denoted $J^{c,\pi}$, where $J^{c,\pi}(s) = \infty$ iff $\pi$ is not $s$-proper. The **optimal value** for a state $s$ under $c$ is given by $J^{c,*}(s) := \inf_{\pi \in \Pi(s)} J^{c,\pi}(s)$. An $s$-proper policy $\pi$ is **optimal** for $s$ under $c$ if $J^{c,\pi}(s) = J^{c,*}(s)$. When $c$ is omitted, we refer to the task's cost function.

## 2.2 Classical Planning & Plan-Space Explanations

Classical planning is special case of probabilistic planning where all actions have a single outcome. For brevity, we denote the unique resulting state of applying $a$ in a state $s$ by $s\llbracket a \rrbracket$, and extend this to action sequences in the obvious manner. Given a state $s$, an action sequence $a_1, \dots, a_n$ is called an $s$-**plan** if it is applicable in $s$ and $\mathcal{G} \subseteq s\llbracket a_1, \dots, a_n \rrbracket$. The set of all $s_0$-plans, also called **plan space**, is denoted $\Pi(s_0)$.

A **plan property** (Eifler et al. 2020a) $\phi$ can be any predicate on plans representing some abstract behavior inter-

pretable by the user. Eifler et al.'s (2020a) methods consider specifically properties $\phi$ expressible as facts $\langle v_\phi, t_\phi \rangle$ such that $\pi$ satisfies $\phi$ iff $s_0\llbracket \pi \rrbracket[v_\phi] = t_\phi$. Richer properties, such as ones formalized via LTL$_f$ (Giacomo and Vardi 2013), can be compiled into this format (Eifler et al. 2020b).

Let $\Omega \subseteq \mathcal{F}$ be a set of user-provided plan properties. Let $\Phi \subseteq \Omega$ be a subset of properties. $\Phi$ is **solvable** if there is a plan $\pi \in \Pi(s_0)$ such that $\Phi \subseteq s_0\llbracket \pi \rrbracket$. In this case, we also say $\pi$ **satisfies** $\Phi$. $\Phi$ is **unsolvable** if it is not satisfied by any plan. $\Phi$ is a **minimal unsolvable subset (MUS)** if $\Phi$ is unsolvable but every $\Phi' \subset \Phi$ is solvable. $\Phi$ **excludes** another $\Psi \subseteq \Omega$, denoted with $\Phi \Rightarrow_\Pi \neg\Psi$, if all plans $\pi \in \Pi(s_0)$ that satisfy $\Phi$ violate $\Psi$. An exclusion $\Phi \Rightarrow_\Pi \neg\Psi$ is **non-dominated** if there is no exclusion $\Phi' \Rightarrow_\Pi \neg\Psi'$ such that $\Phi' \subseteq \Phi$ and $\Psi' \subseteq \Psi$ while one of the subset relations is proper. $\Phi \Rightarrow_\Pi \neg\Psi$ is **non-rhs-dominated** if no $\Psi' \subset \Psi$ is excluded by $\Phi$. Observe that $\Phi \Rightarrow_\Pi \neg\Psi$ iff $\Phi \cup \Psi$ is unsolvable, and it is non-dominated iff $\Phi \cup \Psi$ is a MUS.

Eifler et al. (2020a) distinguish *local* and *global* explanations. The former assume a given subset of properties $\Phi \subseteq \Omega$, and aim at explaining the subspace of plans which satisfy $\Phi$. The $\Phi$-**explanation** gives the consequences of this plan-space restriction in terms of the set $\{ \Psi \subseteq \Omega \mid \Phi \Rightarrow_\Pi \neg\Psi$ is a non-rhs-dominated exclusion$\}$. In contrast, the **global explanation** provides a view on the entire plan space by means of a directed graph over property sets with an arc from $\Phi$ to $\Psi$ if $\Phi \Rightarrow_\Pi \neg\Psi$ is a non-dominated exclusion relation. Computing either type of explanations boils down to solving **AllMUSes**, i.e., computing the set of all MUSes. For global explanations, this is inherent from the definition. For local explanations, this follows from the fact that the non-rhs-dominated exclusions of any $\Phi$ correspond exactly to AllMUSes in the task with goal $\mathcal{G}' := \mathcal{G} \cup \Phi$.

## 3 Exclusion-Based Explanations for SSPs

### 3.1 Policy Properties

We generalize Eifler et al.'s (2020a) concept of plan properties to SSPs. The exclusion-based explanation framework requires properties to be Boolean predicates. In classical planning, each plan $\pi$ uniquely determines an execution trace along with the resulting outcome state $s_0\llbracket \pi \rrbracket$. This makes the evaluation of every condition on $\pi$'s execution inherently Boolean: either the condition is satisfied by $\pi$ (e.g., $s_0\llbracket \pi \rrbracket[v_\phi] = t_\phi$ for the plan property $\phi$) or it is violated ($s_0\llbracket \pi \rrbracket[v_\phi] \neq t_\phi$). One of the two cases definitely applies. In the probabilistic setting, this is more complicated given that policies no longer induce a unique outcome but instead give rise to probability distributions over potentially even infinitely many possible executions, while conditions on individual executions may be satisfied in some but not all the possible executions. Properties expressing characteristics of policies need to take into account this uncertainty, and therewith need to reason over the *expectation* of the properties' objectives. In the remainder of this paper, we will specifically consider two classes of such *policy properties*, one based on *state reachability* and one based on *cost*.

Reachability policy properties extend Eifler et al.'s (2020a) plan properties as introduced above.

**Definition 1** (Policy Reachability Property). *A **policy reachability property** $\phi$ consists of a variable assignment $G_\phi$ and a lower bound $\alpha_\phi \in (0,1]$. A policy $\pi$ satisfies $\phi$ iff $P^{G_\phi,\pi}(s_0) \geq \alpha_\phi$.*

Cost properties allow the user to explore trade-offs between different cost functions and reachability properties:

**Definition 2** (Policy Cost Properties). *A **policy cost property** $\phi$ consists of a cost function $c_\phi$ and an upper bound $\beta_\phi \in \mathbb{R}_0^+$. A policy $\pi$ satisfies $\phi$ iff $J^{c_\phi,\pi}(s_0) \leq \beta_\phi$.*

As in Eifler et al.'s (2020a) framework, we assume the user specifies the relevant policy properties $\Omega$, where $\Omega = \Omega^R \uplus \Omega^C$ is partitioned into reachability $\Omega^R$ and cost properties $\Omega^C$. For a policy-property subset $\Phi \subseteq \Omega$, we similarly use $\Phi = \Phi^R \uplus \Phi^C$. We explain $\Pi(s_0)$ via the induced property-subset exclusion relationships. To this end, the concepts of *solvable* and *unsolvable* property subsets, property-subset *exclusions*, and *local* and *global* explanations are accordingly extended to policy properties. We slightly adapt the conditions of *minimal unsolvable subset* and *exclusion dominance*, taking into account the policy properties' flexibility in the threshold choice. In general, it can be desirable to have in $\Omega$ multiple policy properties $\phi_1, \ldots, \phi_k$ with the same base objective (i.e., same reachability objective $G_{\phi_i} = G_{\phi_j}$, respectively same cost function $c_{\phi_i} = c_{\phi_j}$) but different threshold values. This is useful, in particular, for exploring the effect of different property *relaxations* on the solvability of other properties.

We say that a reachability property $\phi \in \Omega^R$ is a **relaxation** of another reachability property $\psi \in \Omega^R$, denoted $\phi \sqsubseteq \psi$, if $G_\phi = G_\psi$ and $\alpha_\phi \leq \alpha_\psi$; and, similarly, a cost property $\phi \in \Omega^C$ is a relaxation of another cost property $\psi \in \Omega^C$ ($\phi \sqsubseteq \psi$) if $c_\phi = c_\psi$ and $\beta_\phi \geq \beta_\psi$. A relaxation $\phi \sqsubseteq \psi$ with $\phi \neq \psi$ is called **strict** (denoted $\phi \sqsubset \psi$). A property subset $\Phi$ is a relaxation of another property subset $\Psi$, written $\Phi \sqsubseteq \Psi$ if every $\phi \in \Phi$ relaxes some $\psi \in \Psi$. $\Phi$ is a strict relaxation of $\Psi$ ($\Phi \sqsubset \Psi$), if $\Phi \sqsubseteq \Psi$ and $\Phi \neq \Psi$. To define MUSes and policy-property subset exclusion dominance, we substitute the subset relation by relaxation:

**Definition 3** (Minimal unsolvable policy-property subset). *$\Phi \subseteq \Omega$ is a **MUS** if it is unsolvable but each of its strict relaxations $\Phi' \subseteq \Omega$, i.e., $\Phi' \sqsubset \Phi$, is solvable.*

**Definition 4** (Policy-property exclusion dominance). *A policy-property subset exclusion $\Phi \Rightarrow_\Pi \neg\Psi$ is **non-dominated** if there is no other exclusion $\Phi' \Rightarrow_\Pi \neg\Psi'$ such that $\Phi' \sqsubseteq \Phi$ and $\Psi' \sqsubseteq \Psi$ and at least one of the relaxations is strict. Similarly, $\Phi \Rightarrow_\Pi \neg\Psi$ is **non-rhs-dominated** if there is no $\Psi' \subseteq \Omega$ such that $\Psi' \sqsubset \Psi$ and $\Phi \Rightarrow_\Pi \neg\Psi'$.*

Note that both definitions subsume their original counterparts. This follows directly from the observation that if $\Phi' \subseteq \Phi$ for any two policy-property subsets, then it also holds that $\Phi' \sqsubseteq \Phi$. Vice versa, however, there can be non-minimal unsolvable policy-property subsets $\Phi$ (respectively, dominated exclusions $\Phi' \Rightarrow_\Pi \neg\Psi'$), although every subset of $\Phi$ is solvable (respectively, no exclusion satisfies the subset-based dominance criterion). Also note that it is still the case that an exclusion $\Phi \Rightarrow_\Pi \neg\Psi$ is non-dominated iff the combined property set $\Phi \cup \Psi$ is a MUS, and that

$\Phi \Rightarrow_\Pi \neg\Psi$ is non-rhs-dominated iff every relaxation $\Phi' \subseteq \Omega$ of $\Phi \cup \Psi$ with $\Phi \subseteq \Phi'$ is solvable. Finally, observe that for every non-dominated exclusion $\Phi \Rightarrow_\Pi \neg\Psi$, both $\Phi$ and $\Psi$ must be **unambiguous** in that neither of them can contain two properties $\phi$ and $\phi'$ such that $\phi \sqsubset \phi'$.

## 3.2 Exclusion-Explanation Computation

With the ability to express the satisfaction of plan properties as properties on states, the computation of AllMUSes in classical planning degenerates to standard goal reachability questions for different goal sets. Eifler et al. (2020b) showed that these different questions can be solved even with just a single state-space search, resulting in a method offering similar scalability than that of optimal classical planners. Unfortunately, conditions on the expected behavior of policies can no longer be expressed as properties on states. This has two consequences. First, AllMUSes can no longer be computed via methods similar to the algorithm just sketched. Second, whereas in classical planning, a local $\Phi$-explanation could be computed by solving AllMUSes for a goal-extended task, this is no longer possible for SSPs.

**Property-Subset Lattice Exploration** Algorithm 1 depicts our general algorithm for computing local $\Phi$-explanations, and closely resembles previous algorithms for computing minimal unsolvable goal/constraint subsets (Eifler et al. 2020a; Liffiton et al. 2016) with the difference of using property-subset relaxation in place of the subset relation. At its core, Algorithm 1 conducts an exhaustive exploration of the space of all property subsets, testing the solvability of the combined set $\Phi \cup \Psi$ for each visited $\Psi$. As $\Phi \Rightarrow_\Pi \neg\Psi$ holds iff $\Phi \cup \Psi$ is unsolvable, $\Phi$'s non-rhs-dominated exclusions are given exactly by the non-relaxable $\Psi$ for which the combined set was found unsolvable (line 6). To avoid the complete enumeration of all the property subsets, Algorithm 1 utilizes the relaxation relationships with already tested property subsets. The specific choice of $\Psi$ is not important for correctness. In our implementation, we use MARCO (Liffiton et al. 2016), which chooses $\Psi$ in a way guaranteeing that each $\Psi$ added to *Unsolvable* yields a non-rhs-dominated exclusion. Finally, note that by setting $\Phi = \emptyset$, $\min_{\sqsubseteq}$ *Unsolvable* will be exactly the set of all MUSes.

**Theorem 1.** *Algorithm 1 terminates with the $\Phi$-explanation. If $\Phi = \emptyset$ it returns all MUSes.*

**The Property-Subset Solvability Problem** Algorithm 1 relies on a sub-procedure deciding the property subsets' solvability. Before discussing possible implementations of this sub-procedure, here we define and analyze this problem formally.

**Definition 5** (PSS). *The **property-subset solvability problem** (**PSS**) is that of deciding, given a probabilistic planning task $\tau$ and a non-empty set of policy properties $\Phi$, whether there exists an $s_0$-proper policy $\pi$ that satisfies $\Phi$.*

To analyze its complexity, we distinguish decision problems along three lines: (1) classical and probabilistic planning, (2) whether considering the *factored* planning task description or the *flat* state space as input, and (3) in the probabilistic case, whether to allow *stochastic* policies or

**Algorithm 1:** Generic property-subset lattice exploration for computing the $\Phi$-explanation for a given property subset $\Phi$.

**Input:** Probabilistic planning task $\tau$, set of policy properties $\Omega$, property subset $\Phi \subseteq \Omega$
**Output:** All non-rhs-dominated exclusions of $\Phi$
1: *Solvable* $\leftarrow \emptyset$;
2: *Unsolvable* $\leftarrow \emptyset$;
3: **while** true **do**
4:   Pick some $\Psi \subseteq \Omega \setminus \Phi$ such that
     (i) $\forall \Psi' \in$ *Solvable*: $\Psi \not\sqsubseteq \Psi'$, and
     (ii) $\forall \Psi' \in$ *Unsolvable*: $\Psi' \not\sqsubseteq \Psi$, and
     (iii) $\Psi$ is unambiguous
5:   **if** such a $\Psi$ does not exist **then**
6:     **return** $\min_{\sqsubseteq}$ *Unsolvable*
7:   **end if**
8:   **if** $\Phi \cup \Psi$ is solvable **then**
9:     *Solvable* $\leftarrow$ *Solvable* $\cup \{\Psi\}$;
10:  **else**
11:    *Unsolvable* $\leftarrow$ *Unsolvable* $\cup \{\Psi\}$;
12:  **end if**
13: **end while**

whether to constrain the solution space to the *deterministic* policies. It is well-known that with flat representations, the standard plan/policy existence problems can be decided in polynomial time (e.g., Puterman 1994). The policy type does not make a difference. For factored representations, classical planning becomes PSPACE-COMPLETE (Bylander 1994) and probabilistic-planning EXP-COMPLETE (Littman, Goldsmith, and Mundhenk 1998).

Moving to PSS, the additional constraints imposed by the properties have an impact on the complexity in some but not all cases. PSS for the classical-planning flat case is already NP-COMPLETE, being a generalization of the weight-limited shortest-path problem (Garey and Johnson 1979). For the factored representation, PSS stays in the same complexity class of classical planning due to the ability of compiling the properties into facts. PSS for flat SSPs with stochastic policies can still be solved in time polynomial in the size of the SSP via an extension of the standard SSP linear program (as we will see in the next section). For deterministic policies, however, PSS becomes NP-COMPLETE, given the classical planning result. That PSS for factored SSPs is in EXP follows via the approach for the flat variant. Given that standard SSP policy existence is a special case of PSS, the latter therefore remains EXP-COMPLETE when stochastic policies are allowed. However, as in the flat case, the restriction to deterministic policies again raises the complexity by the "non-deterministic" factor. Therefore, and this is in contrast to classical planning, generating explanations for SSPs becomes computationally more demanding than solving the planning task itself (unless NEXP = EXP).

**Theorem 2.** *For singleton property sets $\Phi$, $|\Phi| = 1$, PSS is* EXP-COMPLETE. *For two properties or more, $|\Phi| \geq 2$, PSS is* NEXP-COMPLETE.

## 4  Solving PSS by Search in the Dual Space

Finding a policy that satisfies a given property subset $\Phi$ boils down to solving the SSP under additional *policy constraints*.

This relates to the class of **constrained SSPs (CSSPs)** (Altman 1999). This section develops an extension of I-DUAL (Trevizan et al. 2016) – the so far only known heuristic-search algorithm for solving CSSPs– to solving PSS. To this end, we start with a brief recap of the CSSPs and I-DUAL, then introduce a mixed-integer linear program (MIP) characterization of PSS, and finally show how to adapt I-DUAL to solve this integer program efficiently.

### 4.1  Background: CSSPs and I-DUAL

CSSPs extend SSPs with a list of additional cost functions $c_1, \ldots, c_k$ and accompanying bounds $\beta_1, \ldots, \beta_k$ on the policies' corresponding expected costs. Unlike in the unconstrained version, optimally solving CSSPs in general requires *stochastic* policies, i.e., ones mapping states to probability distributions over actions. The optimal solutions are characterized exactly by the linear program (LP) depicted in Figure 1 (cf., e.g., Altman 1999).

The basis of the encoding are the *occupation-measure (OM) variables* $X_{s,a}$ which represent the expected number of times action $a$ is to be executed in state $s$. The expected cost of the policy represented by the OM variables is given by the linear combination with the cost function (cf. def. of $cost[c']$). The objective function (C1) asks for minimizing the expected cost under the SSP's primary cost function. The bounds on secondary cost functions map into additional constraints (C3). (C4) bounds the states' *flow*, asserting that the expected numbers of times a non-goal state is entered (the *in-flow*) and exited (the *out-flow*) to be the same. A flow surplus of 1 is inserted at the initial state. (C5) implements the $s_0$-proper policy requirement by asserting that the entire inserted flow sinks at the SSP's goal states eventually.

The LP necessitates building the entire state space $\Theta_\tau$. I-DUAL (Trevizan et al. 2016) aims to avoid this by iteratively solving progressively larger sub-SSPs $\hat{\Theta} = \langle \hat{\mathcal{S}}, \mathcal{A}, \hat{\mathcal{T}}, s_0, \hat{\mathcal{S}}_\mathcal{G} \rangle$ of $\Theta_\tau$. One iteration of I-DUAL consists of solving the CSSP LP for the current sub-SSP and subsequently expanding $\hat{\Theta}$ at the *fringe states* visited by the found solution, where $\hat{\Theta}$'s **fringe** $F \subseteq \hat{\mathcal{S}}$ is the set of non-goal states whose transitions from $\Theta_\tau$ are not present in $\hat{\Theta}$ yet. An optimal CSSP solution is found when all flow is sinking at goal states. To guide the exploration, I-DUAL leverages heuristics to estimate the expected remaining cost-to-goal, for each cost function, depending on the reached fringe states. The estimates become additional summands in the objective respectively the cost constraints. Solution optimality is preserved if the heuristics provide are admissible.

Trevizan, Thiébaux, and Haslum (2017) presented an extension of I-DUAL, baptized I²-DUAL, which integrates the computation of a particular heuristic, the **projection occupation-measure (POM) heuristic** $h^{\text{pom}}$, directly in I-DUAL's LP. This has the advantage over using cost-function individual heuristics in being able to *jointly* consider all CSSP's cost constraints inside the heuristic computation. The blue parts in Figure 1 summarize the integration. In brief, $h^{\text{pom}}$ is based on the *projections* of $\tau$ onto the individual state variables $\mathcal{V}$, where the $v$-projection is the simplified task obtained from $\tau$ by discarding all other variables.

Minimize $cost[c] + cost^{\hat{v}}[c]$                                  (C1)

Subject to

$$\mathsf{X}_{s,a} \geq 0, \qquad\qquad s \in \mathcal{S} \setminus \mathcal{S_G}, a \in \mathcal{A}(s) \quad \text{(C2)}$$
$$cost[c_i] + cost^{\hat{v}}[c_i] \leq \beta_i, \qquad\qquad 1 \leq i \leq k \quad \text{(C3)}$$
$$out[s] - in[s] = [s = s_0], \qquad\qquad s \in \mathcal{S} \setminus \mathcal{S_G} \quad \text{(C4)}$$
$$\textstyle\sum_{s \in \mathcal{S_G}} in[s] = 1 \quad \text{(C5)}$$
$$cost[c'] := \textstyle\sum_{s \in \mathcal{S}, a \in \mathcal{A}(s)} \mathsf{X}_{s,a} c'(a), \qquad c' \in \{c, c_1, \ldots, c_n\}$$
$$out[s] := \textstyle\sum_{a \in \mathcal{A}(s)} \mathsf{X}_{s,a}, \qquad\qquad s \in \mathcal{S}$$
$$in[s] := \textstyle\sum_{s' \in \mathcal{S}, a \in \mathcal{A}(s')} \mathsf{X}_{s',a} \mathcal{T}(s', a, s), \qquad\qquad s \in \mathcal{S}$$
$$\mathsf{X}_{d,a}^v \geq 0, \qquad\qquad v \in \mathcal{V}, d \in \mathcal{D}_v, a \in \mathcal{A} \quad \text{(C6)}$$
$$v\text{-projection OM constraints}, \qquad\qquad v \in \mathcal{V} \quad \text{(C7)}$$
$$\text{constraints } tying\ \mathsf{X}_{\cdot,a}^v \text{ with } \mathsf{X}_{\cdot,a}^{\hat{v}}, \qquad v \in \mathcal{V} \setminus \{\hat{v}\}, a \in \mathcal{A} \quad \text{(C8)}$$

Figure 1: LP encoding of a CSSP. $[\varphi]$ denotes the *Iverson bracket* and evaluates to 1 iff $\varphi$ is satisfied and 0 otherwise. $cost[c']$ is a shorthand for the OM variables' induced expected cost under $c'$; $out[s]$ and $in[s]$ for the flow leaving respectively entering state $s$. It is assumed that $s_0 \notin \mathcal{S_G}$. Parts in blue are $\text{I}^2\text{-DUAL}$ extensions; $\hat{v} \in \mathcal{V}$ is arbitrary.

$h^{\text{pom}}$ combines in a single LP (represented by (C6) – (C8)) the OM-based representation of the projections' SSPs, *tying* together the OM variables between the projections by enforcing that each projection executes every action overall exactly the same number of times. By inserting flow into the projections according to the probabilities of reaching frontier states, the projections' OM variables yield the heuristic summands for (C1) and (C3).

## 4.2 Characterizing PSS as a MIP

Figure 2 builds on Figure 1, modeling PSS as a MIP. CSSPs differ from PSS in two regards: CSSPs (1) consider solely constraints on secondary cost functions; and (2) assume stochastic policies. In the presence of a single goal-reachability objective, i.e., assumption (1), one can assume w.l.o.g. that goal states are *absorbing*, forcing policies to immediately terminate once reaching those states. This assumption allows to exclude in the LP (a) the trivial special case $s_0 \in \mathcal{S_G}$, (b) omit goal-state leaving transitions, setting $out[s_\mathcal{G}] = 0$ implicitly for all goal states $s_\mathcal{G}$, and (c) ignore flow constraints for the goal states.

In contrast, in the presence of multiple reachability constraints, continuing the execution in the SSP's goal states might be necessary for achieving the other reachability objectives. Figure 2 accounts for additional reachability objectives by introducing occupation-measure variables also for goal-leaving transitions. Thus, the flow constraints (C11) need to be defined for all states, while to allow flow sinking at some states, the equality needs to be changed to an upper-bound constraint. As flow may now escape from goal states, (C5) is no longer a sufficient proper policy condition. (C12) in Figure 2 instead ensures that all the flow enters and *resides* in the SSP's goal states eventually. Furthermore, (C12) handles the case $s_0 \in \mathcal{S_G}$ by taking into account the initial flow inserted at $s_0$ if $s_0$ is a goal state. Reachability property can now be encoded as flow-residual constraints (C13).

The second source of complication arises from the fact

Minimize (C1)

Subject to

$$\text{(C2)} \qquad\qquad \text{for } all\ s \in \mathcal{S}, a \in \mathcal{A}(s) \quad \text{(C9)}$$
$$\text{(C3)} \qquad\qquad \text{for each } \phi \in \Phi^C \quad \text{(C10)}$$
$$out[s] - in[s] \leq [s = s_0], \qquad\qquad s \in \mathcal{S} \quad \text{(C11)}$$
$$res[\mathcal{S_G}] + \mathsf{P}_\mathcal{G} = 1 - [s_0 \in \mathcal{S_G}] \quad \text{(C12)}$$
$$res[\mathcal{S}_{G_\phi}] + \mathsf{P}_{G_\phi} \geq \alpha_\phi - [s_0 \in \mathcal{S}_{G_\phi}], \qquad \phi \in \Phi^R \quad \text{(C13)}$$
$$\mathsf{D}_{s,a} K - \mathsf{X}_{s,a} \geq 0, \qquad\qquad s \in \mathcal{S}, a \in \mathcal{A}(s) \quad \text{(C14)}$$
$$\textstyle\sum_{a \in \mathcal{A}(s)} \mathsf{D}_{a,s} + \mathsf{T}_s \leq 1, \qquad\qquad s \in \mathcal{S} \quad \text{(C15)}$$
$$out[s] - in[s] + \mathsf{T}_s \geq [s = s_0], \qquad\qquad s \in \mathcal{S} \quad \text{(C16)}$$
$$\mathsf{D}_{s,a} \in \{0,1\}, \qquad\qquad s \in \mathcal{S}, a \in \mathcal{A}(s) \quad \text{(C17)}$$
$$\mathsf{T}_s \in \{0,1\}, \qquad\qquad s \in \mathcal{S} \quad \text{(C18)}$$
$$res[S] := \textstyle\sum_{s \in S} (in[s] - out[s]), \qquad\qquad S \subseteq \mathcal{S}$$
POM extensions (C6) – (C8) with adaptations as discussed (C19) in the text,
$$\mathsf{P}_G \geq 0, \qquad G \in \{G_\phi \mid \phi \in \Phi^R\} \cup \{\mathcal{G}\} \quad \text{(C20)}$$
$$\mathsf{P}_G \leq res^v[G[v]], \qquad v \in \mathcal{V} \text{ s.t. } G[v] \text{ is defined} \quad \text{(C21)}$$

Figure 2: MIP encoding of PSS for property subset $\Phi = \Phi^R \uplus \Phi^C$. Extends the CSSP encoding from Figure 1; objective and constraints are extended to also take into account occupation-measure variables of goal states. $res[S]$ is a shorthand for the flow residing in the state set $S$. $K \in \mathbb{R}_0^+$ is a large constant. Parts in blue are $\text{I}^2\text{-DUAL}$ extensions.

that PSS talks about deterministic policies, which in the presence of additional policy constraints are no longer equally expressive than stochastic policies, as previously mentioned. We implement that requirement by binary integer *decision* variables $\mathsf{D}_{s,a}$ for each state-action pair. (C14) ensures that $\mathsf{D}_{s,a}$ is set if the corresponding transition is assigned a positive occupation measure, while (C15) insists on setting at most one $\mathsf{D}_{s,a}$ per state. The additional binary integer variables $\mathsf{T}_s$ are explicit indicators of the policy terminating in $s$. These are necessary because with the policy's termination being no longer determined by the satisfaction of the SSP's goal, termination becomes a deliberative policy choice. (C15) makes sure that this choice remains deterministic. (C16) ensures that the termination indicator $\mathsf{T}_s$ is set whenever some flow is sinking in state $s$.

Putting everything together, we conclude:

**Theorem 3.** *Figure 2 is feasible iff $\Phi$ is solvable.*

## 4.3 Solving PSS with Heuristic Search

The principles of I-DUAL can be applied directly to solving PSS. The main change required is, obviously, the substitution of the CSSP LP by the MIP from Figure 2. Some care must be taken to correctly handle the subtle differences between PSS and CSSPs. First, given that goal states can no longer be assumed to be absorbing, goal states must now be included in I-DUAL's fringe (and thus possibly expanded). Second, given that goal states may now be contained in the fringe, we must extend I-DUAL's termination condition testing explicitly that no flow is sinking in the fringe. Lastly, the fringe states must be handled appropriately in PSS's

reachability constraints. This is, we must make sure to *optimistically* (hence upper) bound the expected achievement of the individual reachability properties given the fringe states reached. This can be done, in the same vein as for the cost constraints, with the help of admissible goal-probability heuristics. After these changes, I-DUAL's original correctness arguments can be carried over to show that our adapted variant correctly solves PSS.

To complete the adaptation of $I^2$-DUAL, it only remains to extend the $h^{pom}$ part to deliver the necessary optimistic reachability-probability bounds. The blue parts in Figure 2 highlight the main changes. In summary, we discard $h^{pom}$'s proper policy constraint. $h^{pom}$'s reachability-probability bounds are represented through additional variables $P_G$ for each relevant $G$, cf. (C20), which become additional summands in the proper-policy (C12) and the reachability-property constraints (C13) of the overall MIP. The constraints (C21) synchronize the values of those reachability variables with the actual probability of residing in the projections' $G$-achieving states. That this indeed yields an upper bound on the probability of achieving $G$ from the fringe states follows similarly to the cost bounds from the fact that projections are solution preserving.

**Theorem 4.** *Run* $I^2$-DUAL *or* I-DUAL *with admissible heuristics. If at any point in time, the* MIP *for one of the sub-SSPs* $\hat{\Theta}$ *becomes infeasible, then* $\Phi$ *is unsolvable. If they return* $\pi$*, then* $\pi$ *is* $s_0$*-proper and* $\pi$ *satisfies* $\Phi$*.*

Note that the $h^{pom}$ modifications sketched so far do not ensure that the projection's OM variables resemble a deterministic policy. One might be tempted to enforce determinism within every projection via additional integer variables. This however breaks $h^{pom}$'s admissibility property, because due to the tying of the projections, different actions might need to be applied in a single projection state.

# 5 AllMUSes via MO Optimization

Ignoring the properties' thresholds, one is left with a set of policy metrics whose simultaneous optimization becomes a variant of **multi-objective (MO) SSPs** (Chen, Trevizan, and Thiébaux 2023). In the following, we leverage this connection, showing that the solution to this MOSSP variant contains all the relevant information to decide any PSS.

## 5.1 Background: MOSSPs

Multi-objective SSPs (Roijers and Whiteson 2017; Chen, Trevizan, and Thiébaux 2023) differ from regular SSPs in optimizing a **cost-function vector** $\vec{c} = (c_1 \ldots c_n)^T$ rather than a single cost function. Optimality in the MO setting is defined via a dominance order between real vectors, where for two real vectors $\vec{x} = (x_1 \ldots x_n)^T$ and $\vec{y} = (y_1 \ldots y_n)^T$, $\vec{x}$ **weakly dominates** $\vec{y}$ ($\vec{x} \preceq \vec{y}$) if $x_i \leq y_i$ holds for all $i$. $\vec{x}$ **dominates** $\vec{y}$ ($\vec{x} \prec \vec{y}$) if $\vec{x} \preceq \vec{y}$ and $\vec{x} \neq \vec{y}$. Associating each policy $\pi$ with the vector of expected-cost functions $\vec{J}^{\vec{c},\pi} := (J^{c_1,\pi} \ldots J^{c_n,\pi})^T$, the optimal MOSSP policies are those $s_0$-proper policies $\pi$ where $\vec{J}^{\vec{c},\pi}(s_0)$ is not dominated by $\vec{J}^{\vec{c},\pi'}(s_0)$ for any other $\pi'$. Since $\preceq$ is no longer a total order, different optimal policies can in general have different

value vectors. The **optimal MO-value function** $\mathbb{J}^*$ assigns every state to the set of all these optimal cost vectors. Like constrained SSPs, MOSSPs assume stochastic policies, and there can exist $\vec{J}^* \in \mathbb{J}^*(s)$ that are not achievable by any deterministic policy. $|\mathbb{J}^*(s)|$ is in general infinite but can be represented as the convex hull of a finite coverage set (Roijers and Whiteson 2017).

## 5.2 MOSSPs with Deterministic Policies, and Connection to PSS

To be able to translate the policy property set $\Omega$ into a multi-objective optimization problem, we introduce a slight variation of MOSSPs that supports multiple cost as well as goal objectives. Concretely, let $\vec{c} = (c_1 \ldots c_n)^T$ be a cost-function vector, as before, and let $\vec{G} = (G_1 \ldots G_m)^T$ be a vector of variable assignments. For simplicity's sake, we denote for every policy $\pi$ the combined policy value vector with $\vec{V}^\pi = (J^{c_1,\pi} \ldots J^{c_n,\pi} -P^{G_1,\pi} \ldots -P^{G_2,\pi})^T$ (note the negation of the reachability probabilities). A deterministic policy $\pi \in \Pi(s_0)$ is **deterministic optimal** if $\vec{V}^\pi$ is not dominated by $\vec{V}^{\pi'}$ of any $\pi' \in \Pi(s_0)$. The deterministic-optimal MO-value function is given by $\mathbb{V}^*_d(s) := \min_{\preceq}\{\vec{V}^\pi(s) \mid \pi \in \Pi(s)\}$. Note that $|\mathbb{V}^*_d(s)|$ is always guaranteed to be finite.

Let $\vec{c} = (c_0 \ldots c_n)^T$ and $\vec{G} = (G_1 \ldots G_m)^T$ where $c_0$ denotes the SSP's main cost function, $c_1, \ldots, c_n$ are the cost functions underlying the cost properties $\Omega^C$, and $G_1, \ldots, G_m$ are the goal sets underlying $\Omega^R$. Let $\Phi \subseteq \Omega$ be an unambiguous property subset. We associate with $\beta_\Phi(c_i) := \beta_\phi$ the cost threshold assigned by the property $\phi \in \Phi^C$ with $c_\phi = c_i$ if it exists, and define $\beta_\Phi(c_i) := \infty$ otherwise. $\alpha_\Phi(G_i)$ is defined accordingly. Suppose we are given $\mathbb{V}^*_d(s_0)$ for these objectives. Then

**Theorem 5.** *Let* $\Phi \subseteq \Omega$ *be an unambiguous property subset. Let* $\vec{v}_\Phi := (\infty \beta_\Phi(c_1) \ldots \beta_\Phi(c_n) -\alpha_\Phi(G_1) \ldots -\alpha_\Phi(G_m))^T$*.* $\vec{v}_\Phi$ *is dominated by one of the vectors in* $\mathbb{V}^*_d(s_0)$ *iff* $\Phi$ *is solvable.*

Plugged into Algorithm 1, this yields an alternative approach to computing the exclusions, where all property-subset solvability tests boil down to lookups; in particular no additional MIPs need to be solved. For any $\Phi$, the domination condition on $\vec{v}_\Phi$ can obviously be checked in time linear in the number of entries in $\mathbb{V}^*_d(s_0)$. Therefore, if the size of $\mathbb{V}^*_d(s_0)$ is small compared to the number of possible property subsets, then this approach can be expected to be more effective than solving the requested property subsets via individual planner calls – provided that the overhead of the $\mathbb{V}^*_d(s_0)$ precomputation does not outweigh this benefit.

The decision-problem formulation asking whether $\vec{V}^\pi \in \mathbb{V}^*_d(s_0)$, for a given $\vec{V}^\pi$, is equivalent to the definition of PSS (Definition 5). Hence, as a corollary from Theorem 2, this decision problem is NEXP-COMPLETE, which, unfortunately, excludes the use of existing MOSSP techniques.

## 5.3 Enumerating Non-Dominated Solutions

Let $\vec{c} = (c_1 \ldots c_n)^T$ be the cost-function and $\vec{G} = (G_1 \ldots G_m)^T$ be the goal vector. The non-dominated solu-

Minimize

$$\sum_{i=0}^{n} \omega_{c_i}\, cost[c_i] - \sum_{j=1}^{m} \omega_{G_j}\, res[\mathcal{S}_{G_j}] \qquad (C22)$$
$$+ \textcolor{blue}{\sum_{i=1}^{n} \omega_{c_i}\, cost^{\hat{v}}[c_i] - \sum_{j=1}^{m} \omega_{G_j} \mathsf{P}_{G_j}}$$

Subject to

MIP *from Figure 2 without* (C10) *and* (C13) $\qquad$ (C23)

$$cost[c_i] + \textcolor{blue}{cost^{\hat{v}}[c_i]} - \mathsf{W}_{c_i}^{\pi} K \le J^{c_i,\pi}(s_0) - \varepsilon,$$
$$\pi \in Incumbent, i \in \{1,\dots,n\} \quad (C24)$$
$$res[\mathcal{S}_{G_j}] + \textcolor{blue}{\mathsf{P}_{G_j}} + \mathsf{W}_{G_j}^{\pi} K \ge P^{G_j,\pi}(s_0) + \varepsilon,$$
$$\pi \in Incumbent, j \in \{1,\dots,m\},$$

$$\sum_{i=1}^{n} \mathsf{W}_{c_i}^{\pi} + \sum_{i=1}^{m} \mathsf{W}_{G_i}^{\pi} \le n + m - 1, \ \pi \in Incumbent \quad (C25)$$

$$\mathsf{W}_{c_i}^{\pi} \in \{0,1\}, \qquad \pi \in Incumbent, i \in \{1,\dots,n\}$$
$$\mathsf{W}_{G_j}^{\pi} \in \{0,1\}, \qquad \pi \in Incumbent, j \in \{1,\dots,m\} \quad (C26)$$

Figure 3: MIP for enumerating the non-dominated solution vectors $\mathbb{V}_{\mathsf{d}}^{*}(s_0)$. *Incumbent* denotes the set of policies extracted thus far. $\omega$ is a predefined convex combination of all objectives. $\varepsilon \in (0,\infty]$ is the precision parameter. Blue parts show $\mathrm{I}^2$-DUAL modifications.

tion vectors $\mathbb{V}_{\mathsf{d}}^{*}(s_0)$ can be enumerated by solving $|\mathbb{V}_{\mathsf{d}}^{*}(s_0)|$ MIPs. Figure 3 depicts the encoding for finding a single new non-dominated solution. The MIP is iteratively refined taking into account the set *Incumbent* of policies computed so far. During the course of all iterations, it is guaranteed that (1) only deterministic-optimal policies are ever added to *Incumbent*, and (2) that for each value vector $\vec{V}^{*} \in \mathbb{V}_{\mathsf{d}}^{*}(s_0)$, there is at most one policy $\pi \in Incumbent$ such that $\vec{V}^{\pi}(s_0) = \vec{V}^{*}$. $\mathbb{V}_{\mathsf{d}}^{*}(s_0)$ is found when the MIP becomes infeasible. Since the number of policies satisfying (1) and (2) is exactly $|\mathbb{V}_{\mathsf{d}}^{*}(s_0)|$, termination must happen after the claimed number of iterations.

The encoding is based on Figure 2, dropping the policy-property constraints. The optimization function is a linear scalarization of all objectives, i.e., any choice of weights $\omega > 0$ such that $\sum_{i=1}^{n} \omega_{c_i} + \sum_{j=1}^{m} \omega_{G_j} = 1$. This suffices to guarantee (1), which can be shown straightforwardly by contraposition. The bulk of the MIP deals with ensuring progress in the sense of forbidding finding the same solutions again. This is accomplished through constraints (C24), which require improving over the previously computed policy in at least one objective. The selection of this improving objective is implemented via binary integer *wildcard* variables $\mathsf{W}_{O}^{\pi}$, for each policy $\pi \in Incumbent$ and objective $O$. When set, $\mathsf{W}_{O}^{\pi} = 1$, the MIP solution may perform worse than $\pi$ wrt. $O$. However, due to (C25), it is not possible to wildcard all objectives. In other words, each iteration finds a non-dominated $\hat{\pi}$ whose value vector $\vec{V}^{\hat{\pi}}(s_0)$ differs from all the previous ones; yielding property (2). There is a little caveat, however. To model strict inequality constraints, enforcing strict improvement in at least one objective, (C24) has to include a non-zero $\varepsilon$ summand. Nevertheless:

**Theorem 6.** *There always exists $\varepsilon \in (0,\infty]$ such that the sketched algorithm terminates with $\mathbb{V}_{\mathsf{d}}^{*}(s_0)$.*

In practice, the exact value of $\varepsilon$ is task specific and may be difficult to find. The $\varepsilon$ parameter can be used to control the precision, and therewith size of the computed solution set, at the cost of losing the formal correctness guarantee.

As before, it is possible to leverage the principles of I-DUAL, solving each MIP in multiple iterations, while expanding the state space incrementally. Along the lines, and in addition to our adaptations from the previous section, this requires taking into account the fringe states' optimistic heuristic estimates in the ensure-progress constraints (C24); and, to foster finding non-dominated solutions, in the optimization function (C22). Figure 3 illustrates the changes for $\mathrm{I}^2$-DUAL's $h^{\mathrm{pom}}$ representation. Correctness follows with similar arguments as before.

# 6 Experimental Evaluation

The focus of our experiments is evaluating the feasibility of the proposed explanation architecture. Our implementation is based on Probabilistic Fast Downward (Helmert 2006; Steinmetz, Hoffmann, and Buffet 2016). Code and benchmarks will be made publicly available. All experiments were run on a cluster with Intel Xeon E5-2695v4 CPUs, using runtime and memory limits of 30 minutes and 4 GB.

**Setup** We implemented Algorithm 1 via MARCO (Liffiton et al. 2016), using MiniSAT 2.2 (Eén and Sörensson 2003) as the SAT solver. The LPs/MIPs were solved using CPLEX 22.11. We compare 6 PSS methods: *PSS-MIP* via the MIP encoding of PSS, either *(F)* building and solving the MIP over the full (reachable) state space directly, or *(I)* solving the MIP incrementally via our I-DUAL variant using as heuristic the state-of-the-art canonical PPDB heuristic over all patterns of size 2 (Klößner and Hoffmann 2021; Klößner et al. 2021), or *(Î)* solving the MIP with our $\mathrm{I}^2$-DUAL variant. The other three configurations *(MO)* compute up front the optimal MO value function, using either F, I, or $\mathrm{I}^2$ for enumerating the non-dominated solution vectors. We set $\varepsilon = 0.05$. As a reference, we also experimented with LP relaxations of the PSS-MIP methods, which compute the non-dominated exclusions for the space of stochastic policies. This is not possible for the MO methods as the size of the stochastic MO value function might not be finite. We cannot compare to other MOSSP solvers (e.g., Chen, Trevizan, and Thiébaux 2023) due to the lack of support of goal reachability objectives. As an additional reference, we experiment with a state-of-the-art MaxProb heuristic search configuration using iLAO* (Hansen and Zilberstein 2001) and FRET-$\pi$ (Steinmetz, Hoffmann, and Buffet 2016).

**Benchmarks** Our benchmark set is based on existing and new PPDDL benchmarks. A benchmark instance here consists of a PPDDL task $\tau$, a set of properties $\Omega$, and a property subset $\Phi \subset \Omega$ for computing local explanations. The benchmark set is composed of three parts. We leave $\Phi$ empty except in the first part:

- **OSP** Following Eifler et al. (2020a), we generate "OSP" variants of existing benchmarks from the IPPCs and other sources (Steinmetz, Hoffmann, and Buffet 2016; Klößner et al. 2021) that have more than one goal fact. We use $\Omega$ as the representation of "soft goals" and $\Phi$ to enforce the cost bound. Specifically, for each PPDDL base task $\tau$, we generate for each combination of $\mathsf{p} \in \{0.6, 0.75, 0.9\}$ and $\mathsf{c} \in \{0.25, 0.5, 0.75\}$ a benchmark

instance (1) making the goal of the task empty, (2) each original goal fact $g$ becomes a reachability property $\phi_g$ with $G_{\phi_g} = \{g\}$ and $\alpha_{\phi_g} = \mathsf{p} \cdot P^{\mathcal{G}*}$ where $P^{\mathcal{G}*}$ is the MaxProb of $\tau$; (3) $\Phi = \{\phi\}$ where $\phi$ is the cost property enforcing as bound $\beta_\phi = \mathsf{c} \cdot h^*$ on $\tau$'s cost function, where $h^*$ is the minimal path cost required to reach $\tau$'s goal ($J^*$ cannot be used as some benchmarks have no proper policy).

- **Search and Rescue** An adaptation of Trevizan, Thiébaux, and Haslum's (2017) CSSP benchmark. $m$ cells of an $n$-by-$n$ grid can possibly hold a victim. The agent must navigate through the grid in order to find, and as necessary, rescue the victims, while there are bounds on the total time and fuel consumption. We created 75 instances, randomly generating 5 instances for each combination of $n \in \{5, \ldots, 10\}$ and $m \in \{2, 3, 4\}$. $\Omega$ includes the reachability properties $\phi_x$ requiring the victim from cell $x$ being rescued with probability of 1; and for the time and fuel cost functions, the cost properties $\phi_{c,\mathsf{c}}$ with $\beta_{\phi_{c,\mathsf{c}}} = \mathsf{c} \cdot J^{c,*}$, where $J^{c,*}$ is the minimal expected cost under $c$ of rescuing all victims, and $\mathsf{c} \in \{0.5, 0.75, 1.0, 1.25, 1.5\}$. Note that $\mathsf{c} > 1$ makes sense, because being optimal wrt. one cost function does not mean being optimal wrt. the other cost function.

- **Factory** A use-case of an actual company. Incoming product components must be stashed in first-in-first-out storage racks, and taken out of the racks as requested by the production lines. There is uncertainty about the component requested next. Components can be moved between racks. Ideally, all production lines should be served while using as few racks as possible and moving as few components between the racks as possible. We generated 90 random instances with $\{3, 4, 5, 6\}$ incoming components, $\{2, 3\}$ available racks, and $\{2, 3\}$ production lines. We encode the three different preferences as properties: the probability $\mathsf{p}$ of successfully serving all production lines, considering $\mathsf{p} \in \{0.5, 0.6, \ldots, 1.0\}$; a reachability property $\phi_n$ where $G_{\phi_n}$ encodes that $n$ racks have not been used and $\alpha_{G_{\phi_n}} = 1$; and bounds on component movements between racks, modeled as the cost property $\phi_m$ where $\beta_{\phi_m} = m$ and $m \in \{1, \ldots, 5\}$.

**Results** The left-hand side of Table 1 displays the coverage results. Comparing the three MIP solving strategies, $\mathsf{I}^2$ has a clear advantage in both PSS-MIP and MO algorithm variants. Although constructing the full MIP was actually often possible, $\mathsf{F}$ can solve only the very smallest instances, and typically timed out already during the first PSS call. The comparison between $\mathsf{I}$ and $\mathsf{I}^2$ shows the advantage of $\mathsf{I}^2$-DUAL's heuristic being able to simultaneously reason across all the property objectives. $\mathsf{I}^2$ considered on average only half as many states than $\mathsf{I}$. Exceptions are Elevators and Factory where $\mathsf{I}^2$-DUAL's heuristic caused a runtime overhead overshadowing that advantage. The PSS-MIP and MO methods perform overall similarly. However, in some domains (notably Se&Re and Factory) the enumeration of $\vec{V}^*$ turned out infeasible due to too many non-dominated but almost indistinguishable value vectors. Comparing the runtime of PSS-MIP and MO using $\mathsf{I}^2$-DUAL, MO is consid-

| | Coverage | | | | | | | | Time (s) | |
|---|---|---|---|---|---|---|---|---|---|---|
| | Reference | | PSS-MIP | | | MO | | | MIP | MO |
| | $P^{\mathcal{G}*}$ | LP | F | I | $\mathsf{I}^2$ | F | I | $\mathsf{I}^2$ | $\mathsf{I}^2$ | $\mathsf{I}^2$ |
| Blocksw (135) | 108 | 66 | 26 | 46 | **50** | 25 | 31 | 49 | 14.7 | 86.8 |
| Elevators (135) | 135 | 82 | 48 | **72** | 69 | 43 | 46 | 47 | 2.9 | 46.5 |
| ExpBlock (126) | 126 | 63 | 2 | 40 | **50** | 4 | 36 | 41 | 17.6 | 11.4 |
| NoMyst (45) | 36 | 19 | 0 | **18** | **18** | 0 | 13 | 13 | 2.1 | 12.5 |
| Random (108) | 72 | 68 | 0 | 46 | 56 | 3 | 52 | **60** | 33.5 | 20.7 |
| Rovers (90) | 90 | 74 | 22 | **63** | **63** | 26 | 40 | 44 | 30.0 | 46.0 |
| Schedule (90) | 45 | 53 | 18 | 27 | 51 | 18 | 27 | **54** | 8.6 | 2.6 |
| TPP (90) | 90 | 35 | 0 | **26** | **26** | 0 | 8 | 12 | 13.0 | 58.0 |
| Zenotrav (63) | 54 | 28 | 0 | 27 | **28** | 0 | 25 | 25 | 3.2 | 16.0 |
| Se&Re (75) | 75 | 67 | 30 | 33 | **36** | 0 | 0 | 0 | | |
| Factory (90) | 90 | 54 | 24 | **55** | 50 | 26 | 16 | 10 | 42.1 | 773.4 |

Table 1: Number of instances where the explanation was computed within the limits. Abbreviations as described in the text; reference values for solving MaxProb $P^{\mathcal{G}*}$ for the original goal $\mathcal{G}$, and the LP relaxation of PSS-MIP via $\mathsf{I}^2$-DUAL. Time averaged over commonly solved instances.

erably slower almost throughout. It should be noted, however, that MO spends all this time on the computation of $\vec{V}^*$. Once computed, the generation of the explanation only takes a split second. This can become an advantage for computing multiple local explanations in a row, which we did not evaluate here. Compared to the references, the theoretical complexity results are partially reflected in the data. Overall, explanations could be computed in only a fraction of the instances feasible for MaxProb. However, PSS-MIP $\mathsf{I}^2$'s close performance to its LP-relaxed counterpart suggests that the reason of this discrepancy already lies in the secondary constraints and objectives rather than the original source of the complexity increase (deterministic policies).

## 7 Conclusion

In the presence of action-outcome uncertainty, characteristics of solution policies are defined by the expectation over the policies' executions. We introduced accordingly policy reachability and cost variants of Eifler et al.'s (2020a) plan properties. The analysis of mutual relationships between user-provided properties can comprehensibly summarize trade-offs in the infeasibly large space of all (global) or selected (local) policies. We showed that under the restriction to deterministic policies, this analysis is however computationally more difficult than the computation of a single solution policy. Our empirical results reflected to an extent this complexity, but also showed that proposed explanation methodology can be feasible. We introduced two algorithm variants identifying the properties' exclusion relationships. While analyzing property combinations individually tends to be more efficient in computing a single explanation, taking the detour via multi-objective optimization can have strengths in an interactive setting where a user might request a series of local explanations. The adaptation of multi-objective heuristics (Geißer et al. 2022) is a promising direction to improve scalability of the proposed methods.

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
