# OpenReview forum: "Explaining the Space of SSP Policies via Policy-Property Dependencies: Complexity, Algorithms, and Relation to Multi-Objective Planning"
_icaps-conference.org/ICAPS/2024/Conference — ICAPS 2024_

### Official Review · Reviewer_fKQf · 2023-12-27

**Significance And Importance:** 2
**Soundness:** 3
**Novelty:** 3
**Clarity:** 3
**Overall Evaluation:** 1
**Confidence:** 3

**Weaknesses:**

1: Minor weaknesses that are easily fixable.

**Contributions Of The Paper:**

This paper discusses the issue of identifying sets of preferences that can be satisfied in stochastic shortest path problems. The preferences considered are whether a fact is made true in the goal state that is reached by the plan, and bounds on the expectation of multidimensional cost consumed by the plan. A framework algorithm is proposed that computes a directed graph of sets of preferences that are unsatisfiable in combination, and an existing algorithm for solving constrained SSPs is modified to be able to solve the subproblems that are induced by this framework. The resulting algorithm is evaluated on a set of benchmarks, where the authors show that the use of the modified algorithm for solving subproblems improves significantly over a baseline that must enumerate the full state space to solve each subproblem.

**Ethical Considerations:**

(1) Not Applicable: The paper does not have any ethical considerations to address

**Nomination For Best Paper:**

No

**Questions For Authors:**

As I understand it, non-deterministic policies might be able to satisfy preferences that deterministic policies are not able to, e.g. by trying an action that has a positive cost in a different cost function if the bound has already been reached for another one. Is this intuition correct? If so, why focus only on deterministic policies? Is this a limitation of the algorithms used to compute the solution or are there other reasons?

Would some sort of mutex computation improve performance by decreasing the sizes of the different sets that need to be considered? e.g. a fluent that is mutex with the goal could be ruled out directly instead of potentially requiring an iteration of the algorithm.

"too many non-dominated but almost indistinguishable value vectors" would some sort of binning scheme be helpful here to discard entries that are epsilon equal to existing values?

typos etc:

"while planners neither justify the reasons behind their choice of a particular policy over others" unclear what 'neither' refers to here, if meant to contrast with previous part of sentence I would say "nor do planners justify..."

"move all the computational burden to a pre-process" --> to a preprocessing step

"Krarup et al.’s (2021) introduced a generic iterative planning framework" awkward citation format, maybe remove possessive 's, as in the next sentence.

Classical planning is *a* special case of probabilistic planning

if the heuristics provide*d* are admissible

"equally expressive than stochastic policies" than --> as

This is, we must make sure --> That is, we must make sure...

line 541 - using commas between entries in the vector would clarify that the terms are separate and there is no term that is the difference of two values.

line 541 - last term should be −P^G_m,\pi, that is, m rather than 2, correct?

"We created 75 instances" I think the numbers given lead to 80 instances? 6 n values * 3 m values * 5 instances

"only half as many states than" than --> as

**Reproducibility:**

3: Authors describe the implementation and domains in sufficient detail.

**Strengths Of The Paper:**

The paper tackles a difficult problem. New benchmarks (including modifications of existing ones) are proposed for evaluation, and existing algorithms are modified in an interesting way to solve them. The paper shows that algorithms with these modifications substantially outperform a more naive approach.

**Weaknesses Of The Paper:**

I would like to see more motivation for why this is an important problem to solve. Including an example would also help in this direction.

The ideas that are introduced are somewhat difficult to follow and visualize. Perhaps this is unavoidable given the complexity of the problem under consideration, but I think having a running example that is referred to with each new concept would be helpful. I would also suggest including more intuitions in the paper whenever possible to make it easy for the reader to follow.

A figure showing an example output for a small problem, or a part of the output, would be helpful.

Several of the concepts discussed in the paper are more readily understood by many researchers as "soft goals", "rewards", or "utilities" but these terms do not appear until much later in the paper if at all. I think making it clearer early in the paper that these are at least very similar concepts to what is being discussed would make the paper easier to follow for a larger audience.

---

> ### Author Rebuttal · Authors · 2024-01-26
>
> We thank the reviewer for the detailed comments. If accepted, we intend to purchase an extra page for including in particular a motivating running example.
>
> Motivation
>
> There are many use cases that require providing explanations as to why a set of objectives are incompatible under probabilistic uncertainty. For instance, consider the problem of providing explanations to a pilot needing to choose between a number of airports for emergency landing, where, similarly as in (Geisser et al. ICAPS 2020), expected landing time and fuel consumption must be balanced under weather uncertainty.
>
> Non-deterministic policies
>
> Your intuition is correct. Consider the MDP G1 <-[a]- I -[b]-> G2 where G1 and G2 satisfy goal conditions 1 and 2 respectively. Deterministic policies can achieve either 1 or 2. Stochastic policies can satisfy both simultaneously, e.g., {G1=0.5, G2=0.5}, by applying a and b with the according probability.
>
> Stochastic policies are undesirable in many applications, in particular safety-critical ones where one needs predictable behavior, or when actions need to be synchronized among multiple entities, such as for aircraft collision avoidance. From a practical perspective, the algorithms don't immediately impose limitations. The presented methods can generate explanations for stochastic policies by relaxing the MIPs to LPs; in fact, PSS for stochastic policies becomes a CSSP, which, as hinted in the paper, are typically solved through LP-based approaches related to ours.
>
> Mutexes:
>
> Indeed, mutex information can rule out property subsets. Properties mutex with the SSP goal can obviously be discarded immediately. For property combinations in general, this is however more difficult, and particularly depends on the specific probability bounds. For example, assume two mutex goal conditions G1 and G2. Then {G1>=0.6, G2>=0.6} is guaranteed to be unsolvable, because a policy can only satisfy this property set if it reaches a state satisfying both G1 and G2 with probability > 0. In contrast, {G1>=0.5,G2>=0.5} might or might not be solvable.
>
> Binning:
>
> This is indeed what is commonly done in MO optimization (see also Figure 3 and the description). The solution set might however still be huge, which is a consequence of the worst-case size being doubly exponential in the size of the task.
>
> Typos:
>
> Thanks for pointing them out. Reg. search-and-rescue, we only have 75 instances; we did not experiment with n=5 contrary to what is stated in the paper.

---

### Official Review · Reviewer_oL9E · 2024-01-23

**Significance And Importance:** 2
**Soundness:** 4
**Novelty:** 3
**Clarity:** 3
**Overall Evaluation:** 2
**Confidence:** 3

**Weaknesses:**

1: Minor weaknesses that are easily fixable.

**Contributions Of The Paper:**

The paper proposes a framework for plan (policy) explanation (XAI research area). It makes the assumption of having as input a set (Omega) of user-provided plan properties objectives to be verified by the system i.e., to verify if a subset of Omega can not be satisfied. The work is an extention of previous works for Classical Planning to deal with SSPs solution explanation. While for Classical Planning the properties were boolean functions, "properties expressing characteristics of SSP policies need to take into account uncertainty, and thus need to reason over the expectation of the properties objectives".  The paper considers two classes of such policy properties objectives: one based on state reachability and one based on (expected) cost.

In sum, the main contributions are:
- The proposal of a new problem: Property-Subset Solvability problem (called PSS) for SSPs.
- A careful and detailed framework of the theoretical concepts over Exclusion-based Explanation theory for SSPs
- Two approaches for solving PSS problems:  (1) through a series of heuristic searches in the Dual Space (an adaptation based on MIP from Trevizan et al. work) and (2) an alternative method, which leverages a connection to multi-objective probabilistic planning to move all the computational burden to a pre-process.

**Ethical Considerations:**

(1) Not Applicable: The paper does not have any ethical considerations to address

**Nomination For Best Paper:**

No

**Questions For Authors:**

1. How difficult could b  for the user to specify the set of policy properties?

2. Line 652: " We cannot compare to other MOSSP solvers (e.g., Chen, Trevizan, and Thiebaux 2023) due to the lack of support of goal reachability objectives."  See section 3 of the referred paper and comment.

3. What possible adaptations can be done in the work of  Geißer et al. 2022, since you claim that this is a promising direction to improve scalability of the proposed method?

**Reproducibility:**

4: Authors promise to release code and domains (whichever apply).

**Strengths Of The Paper:**

- The proposal of a new problem: Property-Subset Solvability problem (called PSS) for SSPs based on the idea of an explanation being the minimal unsolvable subset (of the user-provided policy properties)
- An original method based on MIP as an adaptation of I-Dual (Trevizan et al) for solving this new problem.

**Weaknesses Of The Paper:**

- The paper organization is kind of chaotic. Between sections 3, 4, and 5, the authors interleave contributions and theoretical background, making it difficult for the reader to understand what is novel and what is background theory. Besides, the paper involves several non trivial and not sufficiently explained approaches such as how to make a MIP encoding of PSS for a property subset based on a MIP encoding of SSPs in the dual space (Trevizan et al). Another non trivial approach, and not well explained as well, is how to use iLAO  and FRET-\pi to solve multi-objective SSPs.

- Since the authors has focused on the theoretical aspects of the paper, the Experimentation section feels too summarized and missing discussions about the setups and results.

- Some typos:
    - Lines 91 and 120: the use of calligraphy G notation for a "conjunctive goal" and regular G notation later.
    - Line 157: an explanation for the meaning of "rhs" is missing.
    - Line 170: is the term "AllMUSes" used as an abbreviation? or an algorithm? It is not clear if the authors are talking about an algorithm called "AllMUSes" (that is able to compute the set of all MUSes) or if it is a theoretical concept (like a group of algorithms that computes all MUSes).

---

> ### Author Rebuttal · Authors · 2024-01-26
>
> We thank the reviewer for the valuable feedback.
>
> Clarity:
>
> We chose to have separate background sections to introduce relevant notations locally, where they are actually needed. We will try making the distinction between our own contributions and background material more clear.
>
> The paper write-up is indeed very dense. If accepted, we intend purchasing an additional page to extend the algorithms' descriptions (including the full MIP encoding of PSS if space permits), and to add additional details to the experiments.
>
> Regarding iLAO and FRET-pi, there might be a misunderstanding. This algorithm combination is used only as a reference baseline solving MaxProb, not multi-objective SSPs. We will clarify this in the final version.
>
> Questions:
>
> 1. Specifying policy properties is no more difficult than specifying plan properties. Our tool chain allows a specification in PDDL. Writing them by hand is typically in the scope of domain experts only, but Eifler et al. (IJCAI 2021) have shown that plan properties can also be learned automatically, which should be extensible to SSPs.
>
> 2. The SSP heuristic search algorithms adapted by Chen et al. do not support 0-cost cycles (which cause the Bellman equations to have multiple solutions, and heuristic search to converge to a suboptimal one), and because of this cannot tackle the cost-compilation of MaxProb. For a detailed exposition of this topic, we refer to (Kolobov et al. ICAPS11). The issue carries over to the multi-objective setting (optimizing a single objective is in fact a special case). We will clarify this in the final version.
>
> 3. Contrary to using independent heuristics for each different objective, Geisser et al.'s MO heuristics can identify combinations of values of different objectives to be unsolvable even when the value of each objective is achievable in isolation. In other words, MO heuristics would have a higher pruning potential in i-dual. Furthermore, like for SSPs, using i-dual with separate heuristics can have advantages over i2-dual, as i2-dual's embedding of the heuristic computation into the MIP sometimes causes a prohibitive overhead. How to integrate MO heuristics into i-dual exactly is however absolutely non-trivial. MO heuristics map states to possibly multiple cost vectors, which to encode into i-dual's constraint program requires additional encoding tricks that go beyond the scope of this paper.

---

### Official Review · Reviewer_8e5R · 2024-01-23

**Significance And Importance:** 3
**Soundness:** 3
**Novelty:** 3
**Clarity:** 3
**Overall Evaluation:** 2
**Confidence:** 3

**Weaknesses:**

2: No major or minor weaknesses.

**Contributions Of The Paper:**

The authors propose a new approach to improving explainability in Stochastic Shortest Path (SSP) problems. There has been a lot of recent work in the deterministic setting, but not as much attention has been paid to the probabilistic case. To address this, they make the following contributions:

1) they extend the definition of plan properties (in their case, policy properties) from Eifler et al to the SSP setting.

2) Unlike in the deterministic case (where checking where a set of plan properties holds equates to checking for goal reachability), they show that checking whether there exists a policy \pi satisfying a set of target policy properties (referred to as PSS in the paper) is harder than planning itself.
Intuitively, this is because checking policy properties is a function of not just the final state, but the entire policy.

3) To address the computational challenge, they leverage existing heuristic search algorithms that solve SSPs (I-dual and I^2-dual) to solve the PSS problem

4) they demonstrate empirically that I^2-dual dominates other approaches (including solving the full MIP directly), demonstrating the promise of their approach.

**Ethical Considerations:**

(5) Excellent: The paper comprehensively addresses all of the applicable ethical considerations

**Nomination For Best Paper:**

Yes

**Questions For Authors:**

Could you elaborate on the anytime aspects you hinted at in the abstract?

**Reproducibility:**

1: Difficult to reproduce because of missing detail.

**Strengths Of The Paper:**

The paper was a joy to read. It's a pretty dense paper in terms of content, yet I was able to follow along without too much effort due to how well-organized and well-written the paper is.

The work is non-incremental. They take a challenging problem, provide a common-sense extension of prior work to formalize it, and leverage some non-trivial connections to existing work in CSSPs and MOSSPs to build tractable algorithms.

**Weaknesses Of The Paper:**

As I said above, the paper is quite well-written. However, the use of minimal examples along the way (e.g. a small problem + a possible set of policy properties one may want to check for) could enhance the paper even more.

L18 in the abstract mentions "anytime" algorithm -- however unless I missed it, I didn't see too much attention paid to that aspect in the empirical evaluation. Given the challenging nature of the problem, quantifying in some ways the quality of solutions built along the way would be useful to see. The empirical evaluation only focuses on

---

> ### Author Rebuttal · Authors · 2024-01-26
>
> We thank the reviewer for the helpful comments. If accepted, we intend purchasing an additional page, allowing to include a motivating running example, and offering additional space for improving clarity of the text where pointed out by the other reviewers, respectively, if possible, provide additional experimental results.
>
> Regarding the anytime behavior, as hinted in section 3.2, the anytime aspect is a characteristic of the MARCO algorithm (i.e., specific variant of Algorithm 1). This is in particular an advantage in an online interactive setting or where the full explanation is not feasible to compute. MARCO iterates over the property subsets in such a way that whenever one is found unsolvable, it is guaranteed to be minimally unsolvable. This makes the unsolvable property subsets found at any point in time a subset of the final explanation that can be provided to the user even before the full construction terminated. The MO-based approaches cannot leverage this benefit as they shift the bulk of the computation in front of MARCO. Due to space reasons, we focused in the experimental evaluation on the computation of the complete explanations. If space permits, we try to include additional plots, such as number of MUSes found over time, elucidating the anytime behavior.

---

### Meta-Review · Area_Chair_4jdr · 2024-02-06

**Recommendation:** Accept (Oral)
**Confidence:** 5

**Metareview:**

The paper applies the general idea behind a recent contribution to explainable AI from classical planning to SSPs. The approach from classical planning analysis user-given plan properties and identifies trade-offs between the given properties. The SSP setting is much more complicated. The authors generalize plan properties to policy properties, define and theoretically analyze the corresponding Property-Subset Solvability problem (called PSS) for SSPs, and propose and evaluate two approaches for solving the problem.
The reviewers are happy with the quality, originality and significance of the work. The paper is very dense and not always easy to follow.

Pros:
- highly non-incremental extension to SSPs
- the paper tackles a difficult and interesting problem
- promising approach
- sound and rigorous

Cons:
- the paper is very dense and some technical details like the full MIP encoding have been omitted.
- no examples
- one reviewer did not like the structure of the paper

**Ethical Considerations:**

(1) Not Applicable: The paper does not have any ethical considerations to address